# Microbial Communities in the Fynbos Region of South Africa: What Happens during Woody Alien Plant Invasions

**Karin Jacobs** [1,*] **, Tersia Conradie** [1] **and Shayne Jacobs** [2]

1   Department of Microbiology, Stellenbosch University, Private Bag X1, Matieland, Stellenbosch 7600, South Africa; tcon@sun.ac.za

2   Department of Conservation Ecology and Entomology, Stellenbosch University, Private Bag X1, Matieland, Stellenbosch 7600, South Africa; sjacobs@sun.ac.za

*   Correspondence: kj@sun.ac.za; Tel.: +27-21-808-5806

**Abstract:** The Cape Floristic Region (CFR) is globally known for its plant biodiversity, and its flora is commonly referred to as fynbos. At the same time, this area is under severe pressure from urbanization, agricultural expansion and the threat of invasive alien plants. *Acacia*, *Eucalyptus* and *Pinus* are the common invasive alien plants found across the biome and considerable time, effort and resources are put into the removal of invasive alien plants and the rehabilitation of native vegetation. Several studies have shown that invasion not only affects the composition of plant species, but also has a profound effect on the soil chemistry and microbial populations. Over the last few years, a number of studies have shown that the microbial populations of the CFR are unique to the area, and harbour many endemic species. The extent of the role they play in the invasion process is, however, still unclear. This review aims to provide an insight into the current knowledge on the different microbial populations from this system, and speculate what their role might be during invasion. More importantly, it places a spotlight on the lack of information about this process.

**Keywords:** fynbos; soil; *Acacia* spp.; *Eucalyptus* spp.; oligotrophic; microbial

## 1. Introduction (to the Fynbos Biome)

The Cape Floristic Region (CFR), also referred to as the core Cape Floristic subregion, at the southern end of the African continent is considered to be one of the most diverse biomes in the world [1–3], with around 9300 described plant species, of which 67% are endemic to the region [3–9] (Figure 1). According to the Cape Town Biodiversity report [9], it is one of 35 global plant biodiversity hotspots. Reflected in this diversity is that of the microbial communities [10–16]. The CFR is characterised by different vegetation types, but can broadly be summarised as Sand fynbos, Alluvium fynbos, Granite fynbos, Sandstone fynbos, Shale fynbos, Renosterveld, Cape flats Dune strandveld, Cape Seashore Vegetation, and the Southern Afro-temperate Forest [6]. The word fynbos refers to the vegetation and originates from the Dutch word meaning "fine bush". Despite the fact that this area is of great ecological value, it also has a major metropole (Cape Town City), which leaves a significant footprint (Figure 1). From their biodiversity report, the City of Cape Town has shown that the different vegetation types within the city limits range from poorly protected to well protected, although more than half are critically endangered [9].

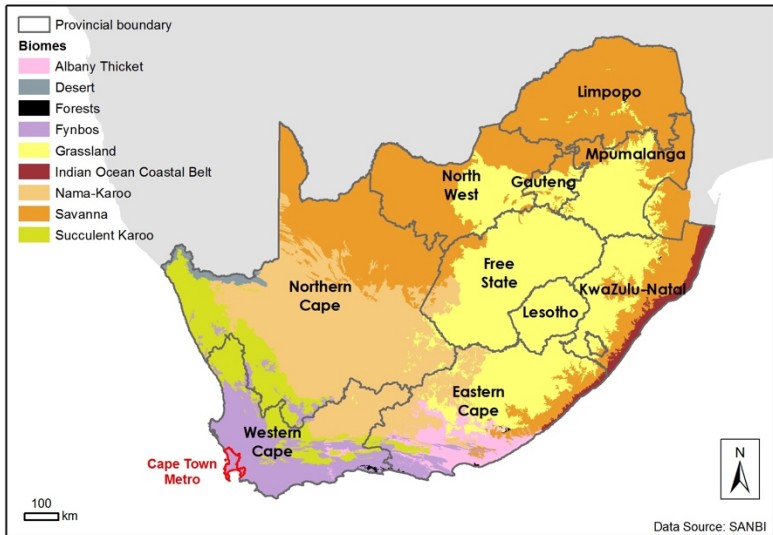

**Figure 1.** Different biomes in South Africa. The Cape Floristic Region (CFR) or fynbos is indicated by the purple areas in the Western part of the country [17]. (Reproduced with permission from [9]).

The loss of fynbos vegetation is mainly due to land use practices that include agriculture and forestry, the expansion of urban areas, and the displacement of native plants through alien plant invasion [5,8]. There is a close link between the above ground and below ground communities. This loss of diversity in the above ground plants are mimicked in the microbial community changes [18]. The fact that many of the vegetation types are critically endangered is troubling and 49 plant species are already confirmed to be extinct [9], and many more are close to extinction within the conservation area (Figure 2). Based on limited previous studies, one can assume that the same is true for the microbial communities, but in the absence of surveys, this can only be estimated.

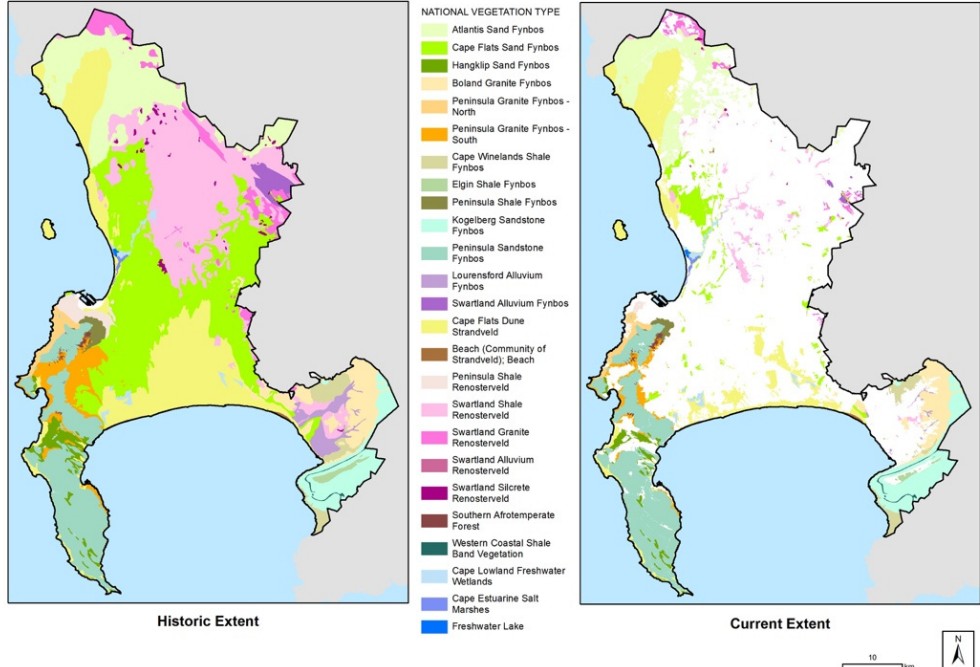

**Figure 2.** Different vegetation types within the fynbos biome showing the historical distribution and remaining patches on the left in 2017 [19]. (Reproduced with permission from [9]).

## 2. Plant–Soil–Microbial Interactions

The high plant diversity is likely the result of a relatively stable climate during the Cenozoic period after rapid changes in the Miocene, which resulted in higher plant diversity [20,21]. This is similar to other Mediterranean ecosystems, where the high diversity in plant species is linked to greater environmental stability [21]. A sixth of all plant species in South Africa is found in the fynbos biome, which covers only 4–6% of the land area [9]. The fynbos biome is mostly known for the iconic *Protea* spp., and 90% of this genus is endemic to the fynbos biome (Figure 3). This is also true for the Ericaceae, and the majority of the diversity in this group can be found in the fynbos biome [22]. Fynbos soils are naturally acidic, and low in nitrogen (N) and phosphorus (P) [23]. Plants, together with a suite of microorganisms, are adapted to this oligotrophic environment through increased plant biomass and biochemical partitioning to roots, cluster roots, exudation of organic acids and phosphatase enzymes for phosphorous acquisition [24,25]. In a semiarid Mediterranean ecosystem, microbial communities were found to be resilient, although they were subjected to strong seasonal shifts and prolonged droughts [26]. This could potentially be true for the fynbos biome, although this has not been tested.

A complex ecosystem of bacteria, fungi, protists and animals inhabits the soil [27], and mostly exhibit positive ecological interactions that promote plant growth. In order to make nutrients such as nitrogen and phosphorus, available to plants, both plants and microorganisms excrete enzymes to cleave nitrogen and phosphorus from organic forms [28,29]. Nitrogen limitations of soils are also overcome by the fixing of atmospheric nitrogen by various bacterial taxa associated with leguminous plants, as well as a number of free-living bacteria [30]. The interaction between plant roots, soil and microorganisms is most intense in the rhizosphere and plant exudates play a significant role in structuring the microbial communities [31]. Within the rhizosphere, there are increased concentrations of phosphatases as well as acidification, enhancing the uptake for both nitrogen and phosphorus. Phosphorus uptake is also enhanced by the activity of mycorrhizae in the root system. Microorganisms in the rhizosphere react with multiple metabolites released by plant roots, where they have mutualistic relationships that favor plant growth, change nutrient dynamics, and also alter the plants vulnerability to heavy metals, abiotic stress, and diseases [32]. In return, plants release compounds in the form of root exudates, creating a unique environment in the rhizosphere and providing microbes with essential components such as sugars, amino acids, flavonoids, aliphatic acids, proteins and fatty acids [28,33]. In some cases, plants can also use their exudates to suppress the growth of microorganisms to reduce the competition for nutrients in low concentrations [31]. Although plants can exert a strong selection pressure on the communities in the rhizosphere, ultimately, the inherent soil composition and nutrient availability is critical to the stability of the system [34]. Spatial variation of these nutrients, as well as soil pH, is important in determining the spatial distribution of fynbos plant species [35].



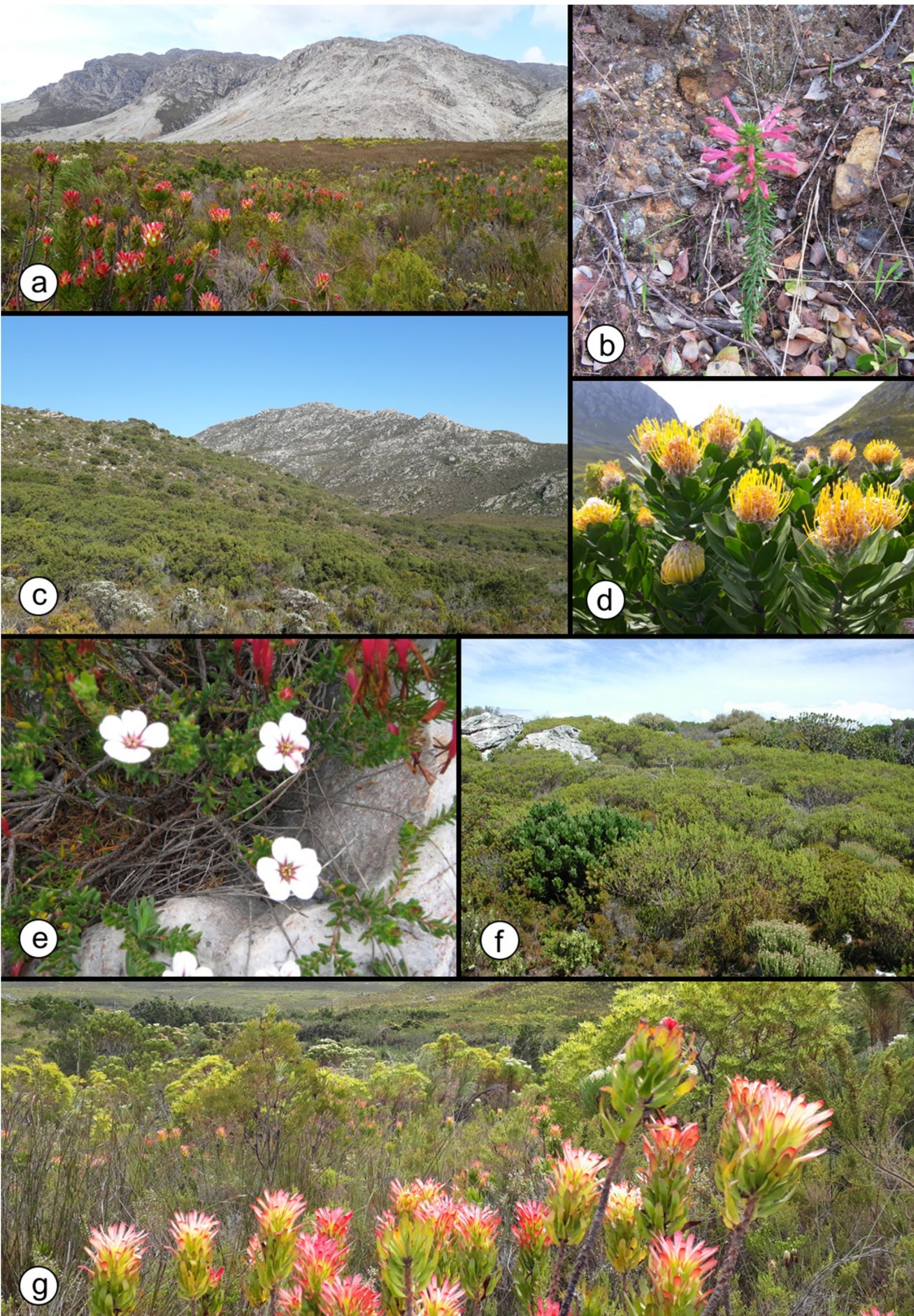

**Figure 3.** Examples of fynbos plants and vegetation. (**a**) Kogelberg Sandstone fynbos (photo: T. Conradie). (**b**) *Erica* flowers. (photo: K. Jacobs). (**c**) Elgin Sandstone near Pringle Bay (photo: T. Conradie). (**d**) *Leucospermum* (Pincushion) flowers (photo: T. Conradie). (**e**) *Coleonema pulchellum* (Confetti bush) flowers (photo: K. Jacobs), (**f**) Peninsula sandstone fynbos at Cape point (photo: K. Jacobs). (**g**) *Protea* flowers in bloom (photo: T. Conradie).

### 3. Microorganisms in Natural Fynbos

Microbial fynbos communities tend to be transient and the compositions of these communities are strongly affected by abiotic factors, particularly pH, moisture and the availability of nutrients [11,36,37]. This was evident from several studies [10,11,38] that showed a clear shift in the bacterial community structure associated with the rhizosphere of native fynbos species between wet and dry seasons, while another study [39] also showed a strong effect of soil pH on microbial structure. Many microbial species also have the ability to switch between an active and dormant state, making them more resilient and resistant to stress in the environment [37,38], and this is most likely the case with species in the fynbos. However, it is important to note that plant–soil interactions are not simple, and usually involve a consortium of organisms with many complex interactions that influence the outcome [31]. From the limited number of studies looking at the microbial species associated with native fynbos plants, it is evident that there is a large number of novel species in this environment, although the majority of these remain undescribed and their function unknown.

### 3.1. Fungi

Mycorrhizal associations occur in 80% of land plants [40] and play an important role in the physiology and health of both the host plant and the fungal symbiont. The arbuscular mycorrhizal fungi (AMF) are mostly believed to be the largest group of plant symbionts and around 62% of fynbos plants are able to form associations with AMF, while 9% are ericoid mycorrhizal, 2% have orchid mycorrhizae, mostly associated with *Disa* spp., and 27% are non-mycorrhizal [41]. These estimates vary depending on the vegetation type and invasion status. Lower colonisation rates of these mycorrhizal fungi (42% of plant species) have been reported from some areas [42,43] as a result of disturbances. For example, the colonisation rates of AMF decreased after fire events, but was able to increase again after two growing seasons [43].

AMF have co-evolved with plants throughout evolution [40,44]. Although AMF is known to assist plants in phosphorus and nitrogen acquisition [41,45], they are absent from proteaceous plants in low nutrient soils [41], even though they can form AMF associations with these plants in soils with a higher nutrient content [41]. One study showed that the cost of maintaining mycorrhizal associations increases with plant density in some fynbos plants [46]. The long-term relationship of plants and AMF allowed plants to adapt to a range of environmental changes, albeit slowly [44]. Arbuscular mycorrhizae are formed by relatively few species of fungi, all of which are members of the phylum, Glomeromycota. Although there are a number of studies investigating the presence of AMF in the fynbos, very little is known about their diversity. Known species from the genera *Achaeospora* and *Glomus* have been found to be associated with plants from the fynbos [45,47].

Not all root associated fungi belong to the Glomeromycota. The Ericaceae is a large group of flowering plants in the fynbos biome that forms mycorrhizal (ERM) associations with non-vesicular-arbuscular species. The evolution of ERM and the diversification of the fungi involved in mycorrhizal associations were instrumental in the spread of plants into new and different habitats, especially those with nutrient-poor soils [40,44,48,49]. Of the approximately 840 known *Erica* spp., about 680 are found within the fynbos biome, and this profusion of ericaceous host plants, together with the oligotrophic nature of the soil, suggests that ericoid mycorrhizal fungi could be abundant in fynbos soils [50–52]. This intense speciation has led to the suggestion that, if ericoid mycorrhizae are required by ericaceous plants for growth and survival in harsh edaphic conditions, then the fungi forming ericoid mycorrhizae must be abundant in fynbos soils [50]. Although only two surveys of the mycorrhizal status of fynbos plants have been completed, all *Erica* plants sampled did have ericoid mycorrhizal associations [41,50], showing that these associations, and therefore the fungi forming ericoid mycorrhiza, are common and widespread in the fynbos region. However, neither of these studies identified the endophytes involved in these associations. Robinson was unable to identify his isolates as they remained sterile in culture (although one isolate was tentatively identified as *Hypoxylon* sp.). Ericoid mycorrhizal fungi from *Erica* plants growing in the fynbos region were used to

characterize the dual phosphate uptake system of ericoid mycorrhizal fungi, however, none of the isolates from these studies were identified [41,50,53–56].

Ericoid mycorrhizae are the most common mycorrhizal type formed in the roots of members of the Ericales. These plants have unique root systems lacking root hairs. Instead, they have ephemeral hair roots: fine, reduced root structures that are most active during moist seasons [57,58]. Although their root systems are dominated by ericoid mycorrhizae, dark septate endophytes and other ectomycorrhizae are also found [59]. Yet, to date, little research has been undertaken on the presence of these fungi in the fynbos biome [41,60,61]. Ericoid mycorrhizae are distinguished from other mycorrhizal types by the distinct hyphal coils that form in the epidermal cells of colonized hair roots. Although adjacent cells may be colonized, each cell is penetrated separately by hyphae growing alongside the root and no lateral colonization occurs [57,58]. Therefore, it is possible for different fungal species to form ericoid mycorrhizal coils in adjacent epidermal cells. Furthermore, no Hartig net or mantle are formed, and arbuscules are not present [62].

In ericoid mycorrhizal associations, the fungal symbiont sequesters nitrogen and phosphate for the host plant and protects the host from high levels of organic acids in the soil in exchange for carbon [57,58,63]. This allows ericaceous plants to grow in habitats with acidic and nutrient-poor soils, such as heathlands and sub-alpine forests. It is thought that the ability of ericaceous plants to thrive in these unfavourable soil conditions is the result of their ericoid mycorrhizal (ECM) associations [62]. The diversity of habitats in which ericaceous plants are found suggests that the ericoid mycorrhizal association fulfils a range of functions, depending on the symbiotic partners involved and the environmental conditions in which the association forms [57]. From an unpublished study, the dominant fungal genus isolated from the common *Erica mammosa* (nine-pin heath), was *Oidiodendron*, while another two representatives were identified as *Cryptosporiopsis ericae* and *Cryptosporiopsis brunnea*. Based on a molecular study of fynbos soils, the common ECM species, *Pezizella ericae*, *Meliniomyces bicolor* and *Meliniomyces vraolstadiae,* have been detected in bulk soil samples from different sites in the fynbos [47] (Figure 4), and most likely form associations with *Erica* spp. in the fynbos biome.

There are many other fungal species associated with both the above and belowground environment of native fynbos biomes. Over the last six decades, various studies reported unique and diverse fungal species from this environment. Although many of these studies are focused on single or only a few plants or habitats, they have contributed to our knowledge of fungi in this biome. Crous et al. [64] estimated that there could be around 171,500 fungal species associated with plants in the fynbos. This is most likely an underestimate as a 1:7 plant to fungus ratio was used, and this estimate does not include those species isolated from soil or insects. No estimate for bacterial diversity in the fynbos are available to date.

Most fungal studies focused on the above ground parts of fynbos plants [65–73] or with obvious disease symptoms [68,74–80]. It is interesting to note that many species are unique to the fynbos, and appear to be endemic [12–15,81–83]. Roets et al. [84] further showed that many of the above ground plants show seasonal trends in their colonisation pattern, as well as host specificity. This further strengthens the hypothesis that the high diversity in plant species is reflected in the fungal populations.

Species of the genera *Penicillium* and *Talaromyces*, are surprisingly well represented in the fynbos [12–15]. These groups have not been well documented [70,85] until recently when a series of studies showed these to be the dominant genera in the fynbos soil biome [12,13,15,47,86] and associated with the above ground structures of various proteaceous plants [12–15,82,83]. The number of species from this area is unexpectedly high with more than 80 species recovered of which almost half are novel [12–15,80,81] (Figure 4). Despite this significant addition to the records of fungi from fynbos, very little is known about the role they play.

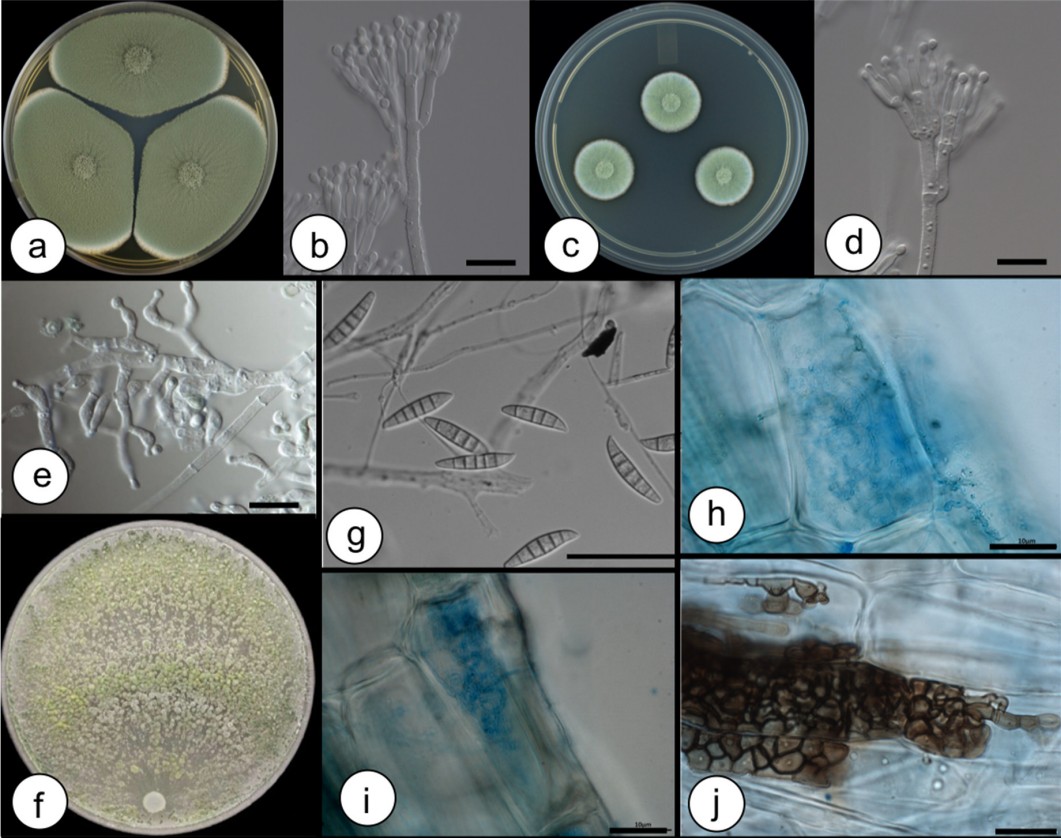

**Figure 4.** Examples of fungi isolated from fynbos soils and plants. (**a**,**b**) Colony morphology and conidiophore morphology of *Penicilium crustosum* isolated from fynbos soil (photo: CM Visagie). (**c**,**d**) Colony morphology and conidiophore morphology of *Penicilium pumilaemali* isolated from fynbos soil (photo: CM Visagie). (**e**,**f**) Conidiophore morphology and colony morphology of *Trichoderma saturnispora* isolated from fynbos soil (photo: IH du Plessis). (**g**) Conidia of an undescribed *Fusarium* sp. isolated from Table Mountain (photo: V Bushula). (**h**,**i**) Mycorrhizal structures observed in *Erica mammosa* hair roots (photo: K Wirth). j. Dark septate endophytes observed in *Erica mammosa* hair roots. (photo: K Wirth). (All scale bars = 10 μm).

Very little is also known about the presence of yeasts in the fynbos biome. A basidiomycetous yeast, *Cryptococcus laurentii*, was isolated from the rhizosphere of the indigenous, medicinal plant *Agathosma betulina* (buchu) [87]. This species was able to colonize the root surface and influence the growth of the plant. Another study [88] showed that the yeast populations of fynbos are mostly influenced by moisture. In Mediterranean systems in other parts of the world, it was shown that the length of a dry summer can have an effect on the structure and function of the microbial communities [89]. This is important as the fynbos region experience very dry summers, frequent fires and occasional droughts, which would have a major impact on the microbial community structure and function.

### 3.2. Bacteria

Bacteria in the fynbos biome have not been intensively studied, and the focus has mostly been on characterising bacterial communities and the effect of disturbance on these communities [10,11,19,38]. Most studies found that the overall bacterial diversity associated with fynbos plants was high, and that these communities appear to be selected by the plant species, possibly through several different mechanisms. The commercialisation of native plant species does not appear to have a major effect on the overall community composition, although closer investigation of these communities reveals an impact on selected groups. However, these communities are heavily influenced by seasonal changes,

with a shift in moisture and temperature. This should raise some questions to the effect of climate change in these areas which are expected to become dryer and warmer in the next few decades [90].

It has been suggested that bacterial groups evolved in this biome to be hyper-diverse and in many cases some species are only found in this part of the world as was the case with *Burkholderia* [91,92] and *Paraburkholderia* [93]. This group of bacteria functions as symbionts of plants to acquire nitrogen through fixation [94]. Although the strains isolated from fynbos plants do not show distinct biogeographic distribution patterns, and were not host-specific, *Burkholderia* spp. appears to have nodulation genes that are closely related and only form nodules in the presences of fynbos papilionoid legumes [95]. Numerous studies isolated and characterised species from this bacterial group, including some novel strains [91,93,96]. Most of the strains isolated from fynbos plants are shown to be highly adapted to low pH environments, consistent with the fynbos soils [97].

Actinomycetes are commonly known as soil bacteria, and can be readily detected from fynbos soils [88]. This bacterial group was recently found to also inhabit the infructescences of *Protea* flowers, and appear to be closely associated with the ophiostomatoid fungi that are regularly associated with the mites inhabiting these habitats [98]. These bacteria are assumed to play an important role in the structuring of the microbial communities in the above ground infructescence as they produce an arsenal of anti-bacterial and anti-fungal compounds and would, therefore be able to select for certain species of fungi and bacteria. As fire is a common occurrence in the fynbos biome, unrestricted fires may have a severe impact on the distribution patterns of these species [98].

One bacterial phylum of interest within the fynbos biome is the Acidobacteria. This bacterial group is commonly found in most soil environments and can represent up to 50% of the 16S rRNA gene sequences [99,100] in molecular studies. Acidobacteria prefer oligotrophic soil environments with a low pH [101,102] and should, therefore be ideally suited to the fynbos environment. In the fynbos biome the Acidobacteria has a relative abundance between 4–26% [10,11,16,38,103,104], with a higher relative abundance estimated in the sand fynbos [104]. The Acidobacteria is divided into 15 subdivisions (formerly 26) [105]. In the fynbos biome, subdivisions 1 (Acidobacteriales), 2 and 3 (Solibacterales) are most commonly found [10,16,38,106]. Although the Acidobacteria is one of the most dominant phyla within the fynbos biome, more than 40% of the acidobacterial affiliated sequences generated with 16S amplicon sequencing are unclassified or unknown. This raises more questions than answers when trying to understand why they are so prominent and what their functions are. Recent advances in metagenomic and proteomic research have started to fill this gap. A study on Mediterranean grassland soil, which is similar to fynbos soil, revealed acidobacterial genomes that encode for large enzyme complements for the degradation of complex carbohydrates (CAZy), making them suitable in an environment where nutrient cycles are microbial driven [107]. Subdivisions 1, 2 and 3 had the highest diversity of CAZy enzymes. They also found that at least one member of the Acidobacteria phylum contained 73% of CAZy classes, which include carbohydrate esterase, glycosyl hydrolase, auxiliary activity, and polysaccharide lyase [107]. Many genomes also possess gene that encode for the capacity to use organic and inorganic nitrogen sources, and the ability to scavenge for atmospheric hydrogen (specifically in subdivisions 1 and 3) [102,107–109]. These traits, together with the ability to use diverse carbohydrates, are advantageous in soil environments where nutrient availability is low and fluctuates between seasons.

### 3.3. Viruses

Very little is known about the virome of the fynbos biome. In a limited study [110], it was found that soils from this area have a diverse representation of viral genomes, while the majority of the gene fragments detected belong to the order Caudovirales, and the majority of the sequences represented bacteriophages. This should not be surprising considering the rich bacterial communities in this domain [10,11,17,93]. The role of these phages in nature is generally considered to be maintaining diversity in populations [111], as well as playing an important part in nutrient cycling [112].

## 4. Alien Plant Invasion in the Fynbos Biome

Alien plant invasion poses a major threat to fynbos vegetation. In 1995, the Working for Water (WfW) program introduced the combined application of mechanical removal and chemical control to remove the invasive species and promote the recovery of fynbos vegetation [8,113–115] (Figure 5). The National Environmental Management: Biodiversity Act (Act 10 of 2004) promulgated actions to control alien plant invasion in fynbos to reduce the area of ecosystems under threat, and prevent the extinction of indigenous species. Actions are also designed to prevent the loss and degradation of soil structure, the function of threatened ecosystems as well as to protect regions of high conservation importance. Over ZAR 15 billion ($9 billion) has been spent to control various invasive species [115]. Nationally, as well as locally in the CFR, mechanical removal and subsequent chemical control are widely used to reduce alien plant invasion.

Several *Acacia* spp., *Eucalyptus* spp., and to a lesser extent species of *Pinus* are major threats to the fynbos vegetation, many of which are considered transformer species, i.e., invasive species that alter the character and nature of recipient ecosystems [116]. Invasive species in both the genera *Acacia* and *Eucalyptus* have specific adaptations that allow them to outcompete native species [117–119], which may lead to a significant decline in native species richness [120]. *Acacia* spp. are more widely distributed in fynbos compared to *Eucalyptus* spp. [121], and their litter production is greater [122,123]. Invasive alien plant species from areas with similar climates have stronger impacts in invaded environments [120], which means that invasive alien plants from other Mediterranean ecosystems have a much larger impact on the native fynbos biome. In general, invasion by particularly *Acacia*, has significant impacts, which increase with time, including altered soil properties, change in light regime, production of allelopathic compounds and many more. In addition, *Acacia* spp. generate massive seed banks which can take considerable effort and time to manage [124]. In other Mediterranean ecosystems, invasion by *Acacia dealbata* was shown to have significant effects on the soil chemistry as well as the structuring of microbial communities [125].

Both *Acacia* and *Eucalyptus* spp. produce litter that influence soil and biogeochemical processes, which ultimately alters the ecosystem functioning [47,117,126]. The phosphorus concentration in the leaves and litter of *Acacia* and *Eucalyptus* spp. are similar [122], however, the nitrogen concentration of *Acacia* spp. litter is relatively higher due to the plants' ability to fix atmospheric nitrogen through nodulating bacteria [122]. In riparian zones, *Acacia* litter was enriched in nitrogen, and *A. mearnsii* resorbed phosphorus rather than nitrogen from senescent leaves, mainly due to the abundance of nitrogen rather than phosphorus in foliage of the nitrogen-fixer [127,128]. Litter rich in nitrogen compared to carbon increases the availability of nitrogen in soil through accelerated decomposition and mineralization [128], which increases the abundance and growth rate of *Acacia* spp. [129,130].

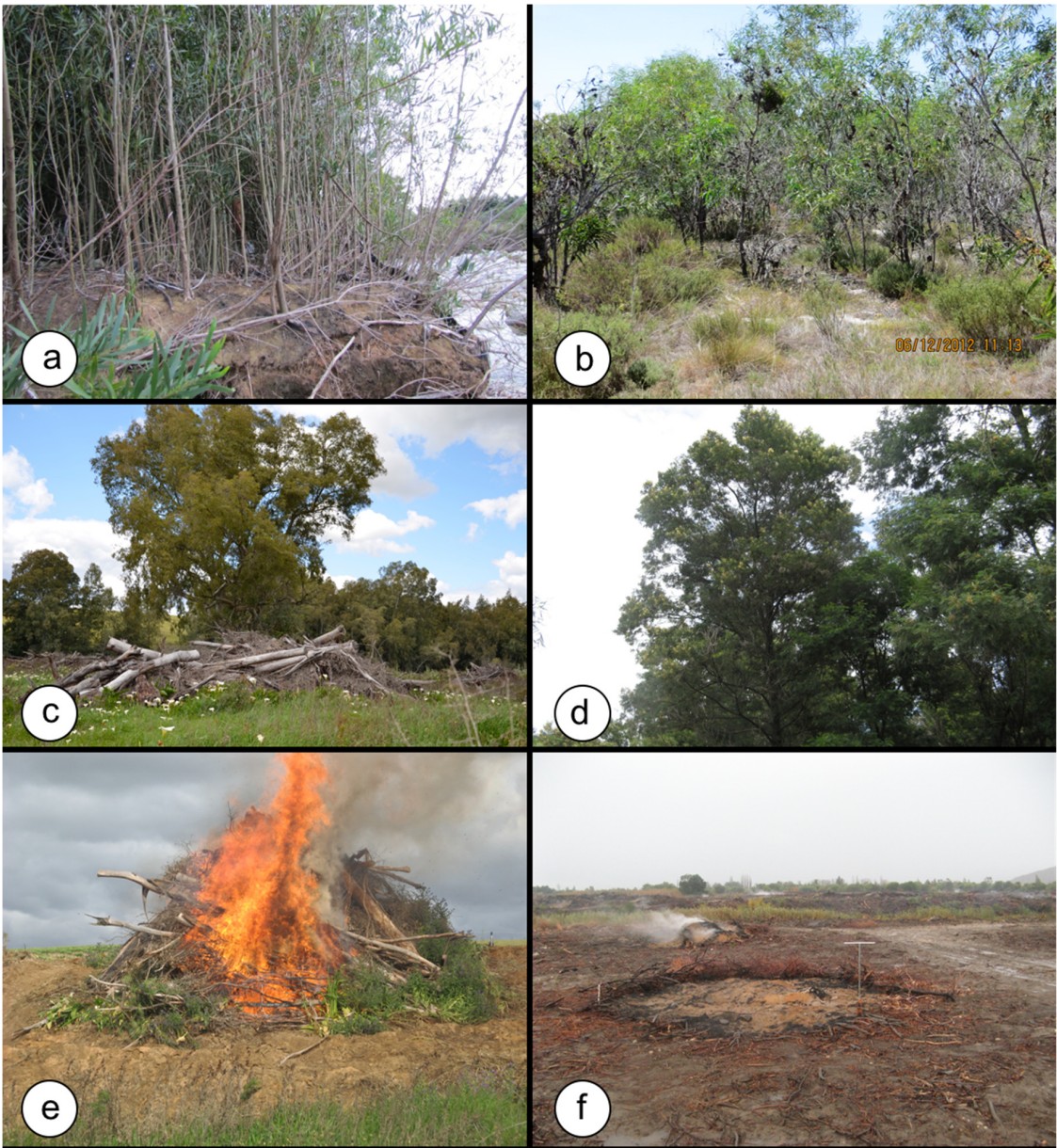

**Figure 5.** Examples of invasive alien plants in the fynbos biome. (**a**) Extensive invasion of riparian zones by *Acacia saligna* (photo: E. Slabbert). (**b**) *A. saligna* growth in the Blaauwberg Nature Reserve park (photo: E. Slabbert). (**c**) Mature *Eucalyptus camaldulensis* growing in a riparian zone along the Berg River at Hermon, with a large pile of alien invasive biomass in the foreground (photo: T. Maubane). (**d**) A large *Acacia mearnsii* tree along the Breede River at Rawsonville (photo: T. Maubane). (**e**) Burning piles of invasive alien plants during slash and burn removal (photo: S. Jacobs). (**f**) The aftermath of burned slash piles, which in some cases may result in permanent burn scars (photo: T. Maubane).

## 5. Changes in Soil Properties

The effect of invasive alien plants on the soil chemistry of fynbos soils can be profound, and by extension should have an influence on the native soil microbial communities. Studies from other Mediterranean ecosystems show that invasive plants significantly change the soil chemistry which leads to an altered microbial community [131]. *A. mearnsii* invasion in the fynbos region drastically alter the soil chemistry with a pronounced effect on the soil microbial communities [11]. Similarly, Yelenik et al. [130] found more than a two-fold increase in soil total available carbon and total nitrogen under *A. saligna* compared to nearby fynbos sites. In riparian environments, a significant

increase in total nitrogen was observed [39], although lower in magnitude compared to the terrestrial sites [130]. This is likely due to loss of soil nitrogen through various hydrological pathways in riverine environments. Available nitrogen also rose significantly in riparian soils, showing the ability of invasive *Acacia* spp. to alter soils and transform ecosystems. In contrast, riparian soils receiving *Eucalyptus* litter were not enriched in total nitrogen [132], similar to the lack of a change in soil N with invasion of *Eucalyptus* spp. in terrestrial soils in a grassy fynbos ecosystem [133]. The alterations in soil nitrogen is thought to be reversible after removal of the nitrogen fixing invasive species, however, several studies showed legacies exist in the form of high levels of available nitrogen well after removal of the invaders [39,130,134]. This may have significant consequences for soil ecology [134], including altering the trajectory of restoration.

The effect of *Acacia* invasion on soil phosphorus is more complex, with no differences in available phosphorus between invaded and natural riparian soils [39]. However, terrestrial soils under *Acacia cyclops* and *A. saligna* stands accumulated higher available phosphorus compared to native fynbos stands [135,136]. For *Eucalyptus* invasion, available phosphorus in terrestrial *Eucalyptus* stands was elevated, which declined upon removal of the invasive trees [133]. Invasive *Acacia*, however, increased acid phosphatase enzyme activity in soils, suggesting that more phosphorus might be available in soils of such invasive stands, though it may be taken up rapidly by the invasive acacias, hence an increase in available phosphorus is not reflected in bulk soil. There is some evidence that showed that soil phosphorus in fynbos riparian soils is linked to bacterial community structure; thus invasion and subsequent changes to soil phosphorus can have significant consequences for soil ecology [11].

Invasion by *Acacia* and *Eucalyptus* spp. into fynbos can also alter soil pH. Invasion of *Eucalyptus* increased soil pH in grassy fynbos [133], while invasion by *Eucalyptus camaldulensis* in riparian zones depressed soil pH by up to 0.3 units [118]. Given that soil pH is strongly linked to soil bacterial community structure, any alteration to soil pH can have major consequences for soil ecology. There are also other impacts, such as an increase in soil hydrophobicity with *Eucalyptus* invasion [137], and an interaction between invasion and fire can alter hydrophobicity in areas where *Acacia* biomass have been stacked and burned [138]. *Eucalyptus* litter also produces allelochemical compounds, which are elevated in soils underneath *Eucalyptus* stands in riparian environments [137]. This advantage that *Eucalyptus* spp. has in its invaded range is suggestive of the novel weapons hypothesis (Table 1).

## 6. The Role of Microorganisms in Invasion

Soil microbial communities can influence invasion, both positively and negatively. It can do so through enhancing or suppressing germination, have positive or negative feedback systems, engage in symbiotic interactions or alternatively influence soil nutrient cycles and soil enzyme concentrations [139–141]. Numerous hypotheses have been put forward to explain or understand the role of microorganisms in the success of invasive plant species [141,142] (Table 1). However, very little research has been undertaken on the role of microorganisms on the success of invasive plants into the fynbos biome.

Introduction of microorganisms into the fynbos biome is not unprecedented. Introducing novel alien pathogens into the fynbos biome has been shown to place plants in this biome under even more pressure and may lead to extensive mortality of indigenous plants [143]. The nodulating bacteria found in fynbos soils and associated with native plants have been studied in some depth. Based on phylogenetic studies, it was shown that the *Bradyrhizobium* spp. in South African fynbos has the same origin as European species [144], although most of the *Bradyrhizobium* species associated with *Acacia* have most likely been co-introduced [145]. *Acacia* spp. also appears to have both introduced some novel rhizobial species into fynbos areas, but also exploited their symbiotic promiscuous association with local rhizobial species. As nodulating bacteria in general are not considered to be good invaders, Rodríguez-Echeverría et al. [146], suggest that broad symbiotic promiscuity and the ability to nodulate at low rhizobial abundance would be beneficial for *Acacia* spp. to invade [146]. Evidence of this can be seen in a number of woody legumes that are effective invaders, [143], and it may well be true for alien

invasive species in the fynbos biome. Distinct bacterial markers are associated with invasion with the numbers of *Bradyrhizobium* showing a rapid increase in invaded areas, alongside a marked decline in other bacterial groups [11].

In the case of non-nodulating species, such as the invasive *Pinus* and *Eucalyptus*, the successful establishment of the tree species relies on its ability to form associations with mycorrhizal symbionts [147]. In the case of *Pinus*, this is a specific interaction, and the ECM is usually co-introduced with the tree. A lack of ECM may in fact be limiting success of *Pinus* spp. in some Mediterranean regions. In the case of the genus *Eucalyptus*, the symbiosis is less specific and *Eucalyptus* spp. can associate with native ECM [140]. In other parts of Africa, non-native ECM fungi were able to become naturalized, and *Eucalyptus* spp. were able to form associations with native ECM fungi [147]. This may be true for *Eucalyptus* and *Pinus* species in the fynbos, but the evidence is still anecdotal.

**Table 1.** Different hypotheses that relate to the contribution of invasion success of non-native plants, by microbial interactions with examples from the fynbos biome (Adapted from [141,142]).

| Hypothesis | Definition | Example | Reference |
|---|---|---|---|
| Enemy release hypothesis | Absence of an antagonist during colonization results in the successful establishment of invaded plant species. | *Sirex noctilio*(woodwasp), together with its symbiotic fungus, *Amylostereum areolatum* infested pine trees in the fynbos region, South Africa decades after establishment of invasive pine as plantations. | [148] |
| | | The rust fungus *Uromycladium tepperianum* was only introduced in 1987 into South Africa as a biocontrol measure on *Acacia*. | [149] |
| Enhanced mutualism hypothesis (novel mutualism) | Invasive plant species associate with native soil mutualists in its introduced ranges which leads to successful invasion. | *Acacia* has been shown to recruit non-specific rhizobia that are native to the fynbos for nodule forming. | [146] |
| | | *Eucalyptus* has been shown to recruit native ectomycorrhizae in other areas of Africa. This has, however, not been shown for species invasive in fynbos, but are likely to occur. | [147] |
| Degraded mutualism hypothesis | The invasion of an area by non-mycorrhizal plants reduces the abundance of arbuscular mycorrhizal (AM) fungi. | The invasion of an area by nonmycorrhizal plants reduces the abundance of arbuscular mycorrhizal fungi (AMF). However, a change in the nutritional status or the absence of important fynbos species such as the Proteaceae may disproportionately select for the re-establishment of AMF-plants, to the detriment of the ECM (Ectomycorrhizal)-plants. | |

**Table 1.** *Cont.*

| Hypothesis | Definition | Example | Reference |
|---|---|---|---|
| Accumulation of local pathogen hypothesis | This suggests that invasive alien plant species gather native soil pathogens that restrict native plant spread and growth. | No evidence. | |
| Novel weapon hypothesis | This postulates that invasive plants possess new biochemical weapons that function as strong allelopathic agents for new plant–soil–microbe interactions. | Slash and burn of Eucalypt during removal of invasive alien plants had a lasting legacy effect on the recovery of native fynbos and changed the soil bacterial communities over an extended time period, most likely as a result of allelochemicals released during decomposition, exacerbated by fire. | [139,150] |

## 7. Effect of Restoration on Microbial Communities

There are many efforts and programmes focused on the restoration of invaded fynbos sites, with various levels of success. These include clearance and removal of the biomass, through various means. In some cases, this is effective, but in most instances, a complex set of interactions are required, such as the re-adjustment of the soil chemistry, reintroduction of native plants and intensive management [116]. It has been suggested that the re-establishment of *Protea* spp., are not influenced as much by the altered soil chemistry as a result of the presence of non-native grass species, but as a result of the presence of beneficial microorganisms in soil [151]. An altered soil microbial community as a result of altered soil chemistry, may, therefore, have a profound effect on the re-establishment of fynbos species in cleared sites, altering the trajectory of recovery, or allowing secondary invasions to take place [152].

The removal of *Acacia* from riparian zones allowed the bacterial communities to recover to one almost resembling the original, over time, with a recovery of native fynbos [11]. One of the most effective ways to remove invasive alien plants is through a slash and burn protocol. This method entails the manual removal of invasive alien plants and stacking the biomass in piles which is then burned (Figure 5). In some cases, the burn will leave a permanent burn scar with little or no recovery of the native fynbos. Burning of *Acacia* and *Eucalyptus* biomass has very different outcomes [150]. In the case of the *Acacia*, the soil bacterial populations will show a shift over the immediate to short term period, while a return to an almost natural fynbos microbial composition is seen over a period of 12 months. However, in the case of *Eucalyptus*, this shift in microbial composition remains after 12 months, with little recovery of the natural vegetation. One of the reasons for this could potentially be the release of toxins and chemicals from *Eucalyptus* into the soil during decomposition [139], exacerbated by burning. Another factor that could alter the trajectory of restoration following clearing is that soil pH can increase by up to two units following burning of biomass piles, possibly due to the release of cations during burning [138]. Differences in soil pH become less pronounced by the end of the first year, which parallels the recovery in soil microbial population structure. Further, following removal of alien invasive trees, herbicides are applied to suppress regrowth. However, the application of herbicide, should it reach soils, could reduce soil pH, showing that soil pH, a major driver of soil microbial diversity, is an important variable that is affected by restoration interventions following clearing and alien species management in riparian and terrestrial areas [138]. This highlights the necessity for a tailored approach to rehabilitation.

## 8. Looking Forward

Given the dearth of information on the diversity of microbial taxa in the CFR, and the threats on plant biodiversity experienced in this globally recognized hotspot, more focused attention is needed on understanding bacterial, fungal and viral biodiversity. This is especially pertinent given the mutualistic relationships between plants and fungi, and the general reliance of endemic fynbos species on particularly fungi and bacteria for survival in this nutrient-poor environment. Although there has been some progress into disentangling the role of microbial communities in the fynbos, there is still much to be done. Research efforts should focus on documenting and characterizing fungal and bacterial species associated with fynbos plants, and not only focus on those causing disease. In addition, high-throughput isolation and molecular methods should be employed to understand the dynamics and function of these communities. Research should focus on restoration practices that will restore and encourage growth of native microbial species, which in turn will enhance the growth of plants in these areas. The identification of keystone microbial species can allow for prioritising restoration projects for greater success, by identifying areas with microbial communities that will enable restoration. However, this will only be possible if efforts to characterise the microbial communities form part of larger conservation efforts and programmes.

## 9. Conclusions

Fynbos plants co-evolved with microbial communities and have adapted to much of the limitations on plant growth by engaging in mutualistic relationships with, e.g., fungi to form ericoid mycorrhiza, and with *Burkholderia* for enhancing nitrogen availability. The existence of other relationships that support the remarkable plant diversity remains unknown, as well as whether this plant diversity is also reflected in a high microbial diversity. The few studies available suggest that this might be the case. Invasion by woody invasive alien plants leads to a major shift in the soil microbial community, which is partially restored when the woody invasive alien plants are removed. A clearer understanding of barriers to restoration of native fynbos plants may be gained through more focused and systematic studies on what happens to soil microbial communities during invasion and after the clearing of invasive alien plants, although the management approach and practices associated with clearing, e.g., use of fire to get rid of biomass, have their own implications for microbial communities. With the double threat of global warming and a growing population, the fynbos biome, like many other ecosystems, is under severe stress. Our ability to respond to these multiple anthropogenic stressors will be severely lacking without a more complete understanding of the microbial communities of the fynbos biome.

**Author Contributions:** Conceptualization, K.J. and S.J.; Writing—Original Draft Preparation, K.J., S.J. and T.C.; Writing—Review and Editing, K.J., S.J. and T.C.; Funding Acquisition, K.J. and S.J. All authors have read and agreed to the published version of the manuscript.

**Funding:** The paper is based on a number of projects funded by the National Research Foundation of South Africa and the Water Research Commission, South Africa.

**Acknowledgments:** The authors of this article would like to thank Charmaine Oxtoby from the City of Cape Town Biodiversity Management Branch, for permission to use Figures 1 and 2, from their report. We also like to acknowledge and thank the reviewers for their valuable comments and suggestions to improve this paper.

**Conflicts of Interest:** The authors declare no conflict of interest.

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
