# Peer review of "Microbial Communities in the Fynbos Region of South Africa: What Happens during Woody Alien Plant Invasions"

_diversity, doi:10.3390/d12060254_

Round 1

Reviewer 1 Report

Dear author,

This review focuses on the regional case study : Cap Floristic Region study. It's original contribution given as CFR presents one of the greatest plant species richness and endemism in the word.

At the read of title and objective of this review, authors wish focus on the plant invasion impact on soil properties and notably soil microorganisms that could modify ecosystem functioning and bring a complexity of ecological restoration. However, review is too descriptive. Scientific issue should be better highlighted. 

Before L 255 is a very long introduction that should be synthesized. Besides objectives and plan of this review should be specified. 

Authors should specify or developp a contempary review on plant-soil microorganism processes of invasive plant species in mediterranean ecosystem to improve understanding degradation, management and restoration of Fynbos ecosystem that an unique ecosystem in the world. 

Authors should better show the link between fundamental knowledge of plant-soil feedback in invasive species and applications in biodiversity conservation and ecological restoration of Fynbos. 

Invasive species impacts on chemical soil properties should be included with the parts on plant-soil microorganisms. 

Conclusion is well written but the output of this review should be specified. Developp this sentence "This review highlights an important gap in our 411 understanding of this ecosystem, and should be prioritised in surveys and policy documents." 

Have good continuation

Regards

Author Response

This review focuses on the regional case study : Cap Floristic Region study. It's original contribution given as CFR presents one of the greatest plant species richness and endemism in the word.

At the read of title and objective of this review, authors wish focus on the plant invasion impact on soil properties and notably soil microorganisms that could modify ecosystem functioning and bring a complexity of ecological restoration. However, review is too descriptive. Scientific issue should be better highlighted. 

>The paper has been written as a review of the current knowledge. While there is significant body of knowledge around plant invasion, very little is known about the role of the microbial communities.  The paper attempt to synthesise the few papers on the microbial communities from this unique environment. 

Before L 255 is a very long introduction that should be synthesized. Besides objectives and plan of this review should be specified. 

>The objective of the study has been included in the abstract.  We would like to argue agains reducing the length of the introduction as it summarise the extent of the microbial communities known from studies done to date.  

Authors should specify or developp a contempary review on plant-soil microorganism processes of invasive plant species in mediterranean ecosystem to improve understanding degradation, management and restoration of Fynbos ecosystem that an unique ecosystem in the world. 

>I do not understand this. However, we have incorporated sections where we compared the CFR with other mediterranean systems. 

Authors should better show the link between fundamental knowledge of plant-soil feedback in invasive species and applications in biodiversity conservation and ecological restoration of Fynbos. 

>There is almost nothing known about the application of microbial knowledge in restoration of Fynbos.  The little we know is included in section 7 on the effect of restoration

Invasive species impacts on chemical soil properties should be included with the parts on plant-soil microorganisms. 

> This is not clear.  I have tried re-arranging the document, but it seems out of place to discuss invasive species when we refer to the soil chemistry of native soils, and their associated microbiology and vegetation. We have, therefore, opted to retain the original placement

Conclusion is well written but the output of this review should be specified. Developp this sentence "This review highlights an important gap in our 411 understanding of this ecosystem, and should be prioritised in surveys and policy documents." 

>We expanded on this in the final paragraph (conclusion)

Reviewer 2 Report

Comments from the Reviewer of the Jacobs et al. manuscript ‘Microbial Communities in the Fynbos Region of South Africa: What Happens during Woody Alien Plant Invasions’ (Diversity)

Jacobs et al. present a review manuscript describing the South African Cape Floristic Region, known as the fynbos, focussing on the impact of invasive plant species and their impact on soil microbiota. This is an interesting review that should be published, and I only have a few minor comments.

The manuscript needs editing to improve grammar, especially in the removal of excess commas, to improve formality. The early part of the review cites references in parentheses, but later on begins to introduce research using author surnames; I would like to see some consistency here and feel that avoiding names and only citing references in parentheses retains formality through the text.

Comments (this is not a comprehensive list of corrections)

  • A comparison should be made early on between the Cape Region and other well-studied regions to put this review in context. Perhaps a table could be used to compare (say) Mediterranean and temperate grassland with the fynbos, indicating plant and soil diversity, and the impact of invasive plant species?
  • Comparisons with other microbial communities and different vegetation types, and different invasive plant species. This review could be improved with more comments linking changes to fynbos microbial communities with other examples.
  • The review could be improved by a section looking forward in terms of interesting ares for research or key management or conservation concerns that need to be addressed in the near future.
  • Line 13. Change to ‘urbanisation and …’.
  • Line 14. Change to ‘biome and …’.
  • Line 15. Change to ‘plants and …’.
  • Line 17. Change to ‘effect on soil chemistry and …’.
  • Line 18. The etymology of the word ‘fynbos’ should be discussed as soon after Line 37 as possible. I understand that it comes from Afrikaans and it is something as a New Zealander I am unfamiliar with; this sort of explanation will be of interest to non-South African readers.
  • Common plant names. It would be good to provide common names for those plants referred to by their species names.
  • Line 39-40. The figure legend should be expanded to draw the reader’s attention to the fynbos indicated in purple.
  • Line 41-41. ‘Urban area’ is not a land use practise and this sentence needs to be corrected for grammar.
  • Line 43. Change to ‘above ground and …’.
  • Lines 50-52. The figure legend needs to be modified to indicate what period the ‘historical extent’ covers and what period the ‘current extent’ covers.
  • Line 54. Change to ‘Cenozoic period’.
  • Line 55. Consider changing to ‘Miocene, which resulted in higher diversity …’.
  • Line 64. Change to ‘exhibit’.
  • Line 67. The statement ‘by various … plants … and … bacteria’ needs to be rewritten as this implies that the legumes are fixing N by themselves, whereas in fact it is the activity of the symbiotic bacteria.
  • Line 72. Delete ‘in the soil’ as technically the soil layer adjacent to the root is known as the rhizosphere.
  • Line 76. Delete ‘the’.
  • Line 84. In order to avoid confusion, I suggest ‘Microbial fynbos communities’ as many other microbial communities are very stable.
  • Line 89. Change to ‘Many microbial species …’.
  • Lines 84-95. It is not really clear in this paragraph why there is an expectation that fynbos microbial communities should differ from those seen in other plant associated communities.
  • Line 104. Change to ‘associations occur in 80% …’.
  • Line 104. Change to ‘[37] and …’.
  • Line 108. Change to ‘spp. and …’.
  • Line 109. Delete ‘may’.
  • Line 113. Change to ‘throughout evolution’.
  • Line 113. Change to ‘Although AMF …’.
  • Line 114. Change to ‘they are absent …’.
  • Line 119. Change to ‘fungi, all of which are members …’.
  • Line 138. Change to ‘sp..’.
  • Line 145. ‘Been done’ is very informal; change to ‘undertaken’.
  • Line 169. Put a space between ‘).E-F’.
  • Line 170. Put a full stop between ‘) G.’.
  • Line 173. Put a space between ‘bars–‘.
  • Line 176. In what sense were the studies ‘sporadic’?
  • Line 179. ‘If one takes’ is very informal; change to ‘as a 1:7 .. ratio was used, …’.
  • Line 192. The phrase ‘is staggering’ is emotive and should be avoided.
  • Line 192. Change to ‘half are novel’.
  • Line 193. Change to ‘Fig.’.
  • Lines 198-199. This statement should be concluded by an indication of why the effect of moisture is relevant to the fynbos.
  • Line 201. Change to ‘have not …’.
  • Line 203. Change to ‘was high’.
  • Line 206. Change to ‘communities reveal …’.
  • Lines 208-209. This sentence needs rewriting for grammar and to provide some further detail; what are the climate change predictions for the Cape region – it is not the case that all areas of the world will see a growth-restricting temperature increase.
  • Line 210. What are ‘hyper-drivers’?
  • Line 211. Burkholderia and Paraburkholderia have been isolated from other places.
  • Line 216. Change to ‘novel strains’.
  • Line 226. Why are the Acidobacteria of particular interest?
  • Line 235-236. This sentence needs rewriting for clarity.
  • Line 237. ‘Done by’ is very informal; change to ‘undertaken’.
  • Line 253. I am not at all sure that this view is commonly held by microbiologists. Transduction is a consequence of errors in replication and phage do not ‘exist’ to limit host numbers. Bacterial (and eukaryote) phage play important roles in nutrient cycling and in maintaining diversity in many systems, including soil and fresh water and marine systems.
  • Line 260. Change to ‘fynbos aimed …’.
  • Lines 260 – 262. This section needs correction – reducing, preventing, and protecting.
  • Line 262. The abbreviation for South African Rand may not be commonly recognised by readers; this should be spelt out in parentheses or the monetary value provided in a more readily recognised international currency.
  • Line 276. Comments referring to references have not been provided elsewhere and should be avoided in this case.
  • Line 285. Change to ‘banks which …’.
  • Line 314. ‘On the other hand’ is very informal; change to ‘However, …’.
  • Line 340. ‘done on’ is very informal; change to ‘undertaken’.
  • Line 342. ‘Unheard of’ is informal; change to ‘unprecedented’.
  • Line 343. Change to ‘biome has …’.
  • Line 344. Change to ‘pressure and …’.
  • Line 351. Correct the accents in the surname.
  • Line 352. Change to ‘abundance would ..’.
  • Table 1. The sections of text and the references need to be aligned neatly and all section of tect in the second and third columns need to conclude with a full stop.
  • Line 383. What is ‘AIP’?
  • Line 408. Change to ‘warming and …’.
  • Line 409. Change to ‘population the …’.

Author Response

Reviewer 2:

Jacobs et al. present a review manuscript describing the South African Cape Floristic Region, known as the fynbos, focussing on the impact of invasive plant species and their impact on soil microbiota. This is an interesting review that should be published, and I only have a few minor comments.

The manuscript needs editing to improve grammar, especially in the removal of excess commas, to improve formality. The early part of the review cites references in parentheses, but later on begins to introduce research using author surnames; I would like to see some consistency here and feel that avoiding names and only citing references in parentheses retains formality through the text.

>We have checked all the intext references for accuracy and removed author names, unless these formed part of the narrative.

>Excessive use of commas were addressed.

  • A comparison should be made early on between the Cape Region and other well-studied regions to put this review in context. Perhaps a table could be used to compare (say) Mediterranean and temperate grassland with the fynbos, indicating plant and soil diversity, and the impact of invasive plant species?

>We have opted to add sections throughout the paper.  These are marked in yellow in the mark-up.  

  • Comparisons with other microbial communities and different vegetation types, and different invasive plant species. This review could be improved with more comments linking changes to fynbos microbial communities with other examples.

>We have used examples from other Mediterranean ecosystems where applicable. 

  • The review could be improved by a section looking forward in terms of interesting ares for research or key management or conservation concerns that need to be addressed in the near future.

> This was added at the end as part of the conclusion.

Line 18. The etymology of the word ‘fynbos’ should be discussed as soon after Line 37 as possible. I understand that it comes from Afrikaans and it is something as a New Zealander I am unfamiliar with; this sort of explanation will be of interest to non-South African readers.

>We added a definition in the first paragraph

Common plant names. It would be good to provide common names for those plants referred to by their species names.

>We have added these where appropriate.

Line 39-40. The figure legend should be expanded to draw the reader’s attention to the fynbos indicated in purple.

>It is included in the legend.

Line 41-41. ‘Urban area’ is not a land use practise and this sentence needs to be corrected for grammar.

>The sentence was corrected

>All of the following corrections were made.  

Editorial: 

Line 13. Change to ‘urbanisation and …’.

Line 14. Change to ‘biome and …’.

Line 15. Change to ‘plants and …’.

Line 17. Change to ‘effect on soil chemistry and …’.

Line 43. Change to ‘above ground and …’.

Lines 50-52. The figure legend needs to be modified to indicate what period the ‘historical extent’ covers and what period the ‘current extent’ covers.

Line 54. Change to ‘Cenozoic period’.

Line 55. Consider changing to ‘Miocene, which resulted in higher diversity …’.

Line 64. Change to ‘exhibit’.

Line 67. The statement ‘by various … plants … and … bacteria’ needs to be rewritten as this implies that the legumes are fixing N by themselves, whereas in fact it is the activity of the symbiotic bacteria.

Line 72. Delete ‘in the soil’ as technically the soil layer adjacent to the root is known as the rhizosphere.

Line 76. Delete ‘the’.

Line 84. In order to avoid confusion, I suggest ‘Microbial fynbos communities’ as many other microbial communities are very stable.

Line 89. Change to ‘Many microbial species …’.

Line 104. Change to ‘associations occur in 80% …’.

Line 104. Change to ‘[37] and …’.

Line 108. Change to ‘spp. and …’.

Line 109. Delete ‘may’.

Line 113. Change to ‘throughout evolution’.

Line 113. Change to ‘Although AMF …’.

Line 114. Change to ‘they are absent …’.

Line 119. Change to ‘fungi, all of which are members …’.

Line 138. Change to ‘sp..’.

Line 145. ‘Been done’ is very informal; change to ‘undertaken’.

Line 169. Put a space between ‘).E-F’.

Line 170. Put a full stop between ‘) G.’.

Line 173. Put a space between ‘bars–‘.

Line 176. In what sense were the studies ‘sporadic’?

Line 179. ‘If one takes’ is very informal; change to ‘as a 1:7 .. ratio was used, …’.

Line 192. The phrase ‘is staggering’ is emotive and should be avoided.

Line 192. Change to ‘half are novel’.

Line 193. Change to ‘Fig.’.

Lines 198-199. This statement should be concluded by an indication of why the effect of moisture is relevant to the fynbos.

Line 201. Change to ‘have not …’.

Line 203. Change to ‘was high’.

Line 206. Change to ‘communities reveal …’.

Line 210. What are ‘hyper-drivers’?

Line 211. Burkholderia and Paraburkholderia have been isolated from other places.

Line 216. Change to ‘novel strains’.

Line 226. Why are the Acidobacteria of particular interest?

Line 235-236. This sentence needs rewriting for clarity.

Line 237. ‘Done by’ is very informal; change to ‘undertaken’.

Line 253. I am not at all sure that this view is commonly held by microbiologists. Transduction is a consequence of errors in replication and phage do not ‘exist’ to limit host numbers. Bacterial (and eukaryote) phage play important roles in nutrient cycling and in maintaining diversity in many systems, including soil and fresh water and marine systems.

Line 260. Change to ‘fynbos aimed …’.

Lines 260 – 262. This section needs correction – reducing, preventing, and protecting.

Line 262. The abbreviation for South African Rand may not be commonly recognised by readers; this should be spelt out in parentheses or the monetary value provided in a more readily recognised international currency.

Line 276. Comments referring to references have not been provided elsewhere and should be avoided in this case.

Line 285. Change to ‘banks which …’.

Line 314. ‘On the other hand’ is very informal; change to ‘However, …’.

Line 340. ‘done on’ is very informal; change to ‘undertaken’.

Line 342. ‘Unheard of’ is informal; change to ‘unprecedented’.

Line 343. Change to ‘biome has …’.

Line 344. Change to ‘pressure and …’.

Line 351. Correct the accents in the surname.

Line 352. Change to ‘abundance would ..’.

Table 1. The sections of text and the references need to be aligned neatly and all section of tect in the second and third columns need to conclude with a full stop.

Line 408. Change to ‘warming and …’.

Line 409. Change to ‘population the …

Line 383. What is ‘AIP’?

>we removed this from the text and used the more excepted term invasive alien species.

Lines 208-209. This sentence needs rewriting for grammar and to provide some further detail; what are the climate change predictions for the Cape region – it is not the case that all areas of the world will see a growth-restricting temperature increase.

>We added a reference and some detail.

Lines 84-95. It is not really clear in this paragraph why there is an expectation that fynbos microbial communities should differ from those seen in other plant associated communities.

>Clarified in the text

Reviewer 3 Report

The manuscript by Jacobs et al. provides a unique discussion topic of how microbial communities in fynbos alter after the invasion by alien plant species. The review is well documented, and the content is very interesting and updated, which could be attractive for Diversity's readers and beneficial for monitoring and overcoming the alien plant invasion and its consequences. I would recommend acceptance of this article after a minor revision. Below you can find some of my comments.

Abstract:

Please avoid using redundant words (e.g., alien invasive plants).

Main Text:

Figure 3: The referring letters in the pictures are small cases, but those described in the figure legend are capitals, please unite them.

Line 106: arbuscular mycorrhizal fungi 

Line 110: mycorrhizal fungi

Line 114: Proteaceous (proteaceous at line 191), should this term be capitalised? What about 'ericaceous host plants' at lines 128, 131, 154, 155, and 157, should it be capitalised too?

Line 133: Erica in italic?

Line 135: ericoid mycorrhizae?

Line 138: Hypoxylon in italic.

Lines 159-162: Confused, please clarify.

Line 162: Change spp. to species

Line 163: Following the binomial system, Meliniomyces vraolstadiae is the determination of a species, which is mentioned for the first time at line 163. Therefore, it should be written in full.

Figure 4: The referring letters in the pictures are small cases, but those described in the figure legend are capitals, please unite them. Line 169: space is missing in front of E-F. Line 170: a full stop is missing behind (photo: IH duPlessis) and Fusarium in italic with sp. Line 172: Delete the full stop behind roots. Line 173: μm, not um, add spaces too.

Line 188: Remove excess comma

Lines 188-191: In the first sentence, the authors mentioned two genera, and in the second sentence, only a genus was mentioned. Please clarify and revise the paragraph.

Lines 210-218: Wrong spelling names of bacteria, please check

Lines 212-213: Please correct English usage

Line 216: this bacterial group

Line 220: this bacterial group

Lines 220-223: Please check your English usage and clarify how could actinobacteria shape the community structures in such niches?

Line 226: this bacterial group

Line 227: Unclear statement, please revise, 16S rRNA gene sequences

Line 231: I don't see this statement 'corresponding to class level' helps; please consider to remove it

Line 233: What do you mean by 'has no taxonomic classification'? No classified taxon or taxonomic member?

Line 234: What sequences? I know what you mean, but what about the others who don't work with bacterial diversity and molecular tools?

Line 249: viral genomes

Figure 5: The referring letters in the pictures are small cases, but those described in the figure legend are capitals, please unite them. Line 268: You can use Asaligna for the second mention

Line 291: nitrogen > N

Line 299: Full-term for Acacia mearnsii is already mentioned at line 292, you can use Amearnsii instead

Line 309: N-fixing invasives

Line 315: Is this first time mentioned Acyclops, if so, please provide the full term.

Line 326: Is this first time mentioned Ecamaldulensis, if so, please provide the full term.

Lines 332-333: The term 'Eucalyptus spp.' is plural; please correct your sentence

Line 363: Eucalyptus in italic

Table 1, page 12:

Ref. 143: Acacia has been shown to recruit non-specific rhizobia that are native to the fynbos for nodule forming 

Ref. 144: Eucalyptus has been shown to recruit native ectomycorrhizae in other areas of Africa. However, this study has not been shown for species invasive in fynbos, but are likely to occur.  

Ref. 45: The invasion of an area by nonmycorrhizal plants reduces the abundance of arbuscular mycorrhizal fungi (AMF)

However, a change in the nutritional status or the absence of important fynbos species such as the Proteaceae may disproportionately select for the re-establishment of AMF-plants, to the detriment of the ECM-plants. Please provide the full term of ECM as the Table should be understandable when it stands alone.

Ref. 137, 145: This postulates that invasive plants possess new biochemical weapons that function as strong allelopathic agents for new plant-soil-microbe interactions.

Please provide the full term of AIP as the Table should be understandable when it stands alone.

Author Response

Reviewer 3:

The manuscript by Jacobs et al. provides a unique discussion topic of how microbial communities in fynbos alter after the invasion by alien plant species. The review is well documented, and the content is very interesting and updated, which could be attractive for Diversity's readers and beneficial for monitoring and overcoming the alien plant invasion and its consequences. I would recommend acceptance of this article after a minor revision. Below you can find some of my comments.

Abstract:

Please avoid using redundant words (e.g., alien invasive plants).

>We replaced this with the more recognised term invasive alien plants. 

>Editorial changes were all made unless indicated otherwise.

Main Text:

Figure 3: The referring letters in the pictures are small cases, but those described in the figure legend are capitals, please unite them.

Line 106: arbuscular mycorrhizal fungi 

Line 110: mycorrhizal fungi

Line 114: Proteaceous (proteaceous at line 191), should this term be capitalised? What about 'ericaceous host plants' at lines 128, 131, 154, 155, and 157, should it be capitalised too?

Line 133: Erica in italic?

Line 138: Hypoxylon in italic.

Line 162: Change spp. to species

Line 163: Following the binomial system, Meliniomyces vraolstadiae is the determination of a species, which is mentioned for the first time at line 163. Therefore, it should be written in full.

Figure 4: The referring letters in the pictures are small cases, but those described in the figure legend are capitals, please unite them. Line 169: space is missing in front of E-F. Line 170: a full stop is missing behind (photo: IH duPlessis) and Fusarium in italic with sp. Line 172: Delete the full stop behind roots. Line 173: μm, not um, add spaces too.

Line 188: Remove excess comma

Lines 210-218: Wrong spelling names of bacteria, please check

Lines 212-213: Please correct English usage

Line 216: this bacterial group

Line 220: this bacterial group

Line 226: this bacterial group

Line 249: viral genomes

Figure 5: The referring letters in the pictures are small cases, but those described in the figure legend are capitals, please unite them. Line 268: You can use Asaligna for the second mention

Line 299: Full-term for Acacia mearnsii is already mentioned at line 292, you can use Amearnsii instead

Line 309: N-fixing invasives

Line 315: Is this first time mentioned Acyclops, if so, please provide the full term.

Line 326: Is this first time mentioned Ecamaldulensis, if so, please provide the full term.

Lines 332-333: The term 'Eucalyptus spp.' is plural; please correct your sentence

Line 363: Eucalyptus in italic

Line 135: ericoid mycorrhizae

> Recognised term for species associated with ericaceous hosts

Lines 159-162: Confused, please clarify.

>Clarified in the text. 

Lines 188-191: In the first sentence, the authors mentioned two genera, and in the second sentence, only a genus was mentioned. Please clarify and revise the paragraph.

>These are closely related genera.  We have correct the use of the names in the text.

Lines 220-223: Please check your English usage and clarify how could actinobacteria shape the community structures in such niches?

>We expanded on this in the text

Line 227: Unclear statement, please revise, 16S rRNA gene sequences

>We expanded on this in the text

Line 231: I don't see this statement 'corresponding to class level' helps; please consider to remove it

>Removed from text

Line 233: What do you mean by 'has no taxonomic classification'? No classified taxon or taxonomic member?

Removed the sentence

Line 234: What sequences? I know what you mean, but what about the others who don't work with bacterial diversity and molecular tools?

Clarified in the text, for a more general audience. 

Line 291: nitrogen > N

>Fixed this throughout the manuscript.

>All suggested changes were made to the table.

Table 1, page 12:

Ref. 143: Acacia has been shown to recruit non-specific rhizobia that are native to the fynbos for nodule forming 

Ref. 144: Eucalyptus has been shown to recruit native ectomycorrhizae in other areas of Africa. However, this study has not been shown for species invasive in fynbos, but are likely to occur.  

Ref. 45: The invasion of an area by nonmycorrhizal plants reduces the abundance of arbuscular mycorrhizal fungi (AMF). 

However, a change in the nutritional status or the absence of important fynbos species such as the Proteaceae may disproportionately select for the re-establishment of AMF-plants, to the detriment of the ECM-plants. Please provide the full term of ECM as the Table should be understandable when it stands alone.

Ref. 137, 145: This postulates that invasive plants possess new biochemical weapons that function as strong allelopathic agents for new plant-soil-microbe interactions.

Please provide the full term of AIP as the Table should be understandable when it stands alone.

>I removed this from the manuscript

Round 2

Reviewer 1 Report

Authors have chosen to organize their ms by biginning the presentation of Fynbos biome. To increase its visibility and scientific impact, I advise to develop the issue on role of microbial communities in functioning of several mediterranean ecosystems and their restoration to conclude on the remarkable Fynbos biome. Objective should be revised by including this major comment. 

This ms is original contribution that is well written. 

I accept the manuscript leaving the choice to the authors to reorganize their manuscript.

Regards